# Unlocking Barriers to Circular Economy: An ISM-Based Approach to Contextualizing Dependencies



**Steffen Foldager Jensen** [1,*], **Jesper Hemdrup Kristensen** [1], **Jonas Nygaard Uhrenholt** [1,2], **Maria Camila Rincón** [1], **Sofie Adamsen** [1] **and Brian Vejrum Waehrens** [1]

1    Center for Industrial Production, Department of Materials and Production, Aalborg University, 9220 Aalborg, Denmark; jhk@mp.aau.dk (J.H.K.); jonasn@mp.aau.dk (J.N.U.); mcrg@mp.aau.dk (M.C.R.); sofiea@mp.aau.dk (S.A.); bvw@mp.aau.dk (B.V.W.)

2    Technology & Business Development, University College of Northern Denmark, 9200 Aalborg, Denmark

\*    Correspondence: steffenfj@mp.aau.dk; Tel.: +45-40467499

**Abstract:** Despite it being imperative to a sustainable development, a circular economy remains scarcely adopted by companies. Barriers towards this are extensively explored yet with little focus on their mutual dependencies. Neglecting dependencies is argued to cause suboptimization and lead to unsuccessful circular projects. To counter this and strengthen companies in assessing dependencies among context-dependent barriers towards a circular transition, this study proposes a practice-oriented approach based on an interpretive structural modelling methodology. This is validated through a case study with a Danish mechatronics manufacturer with which fourteen semi-structured interviews, a survey, and a workshop were conducted. Findings reveal an interwoven network of barriers with numerous chain mechanisms across managerial, market-related, financial, technical, and regulative aspects, which underpins the need to approach the circular transition systemically. Furthermore, the study demonstrates the ability of the methodology to facilitate discussions and assist industrial practitioners, both on a strategic and operational level, in systematically untangling the complex interrelations to identify root causes for inertia and prioritize mitigation measures.

**Keywords:** sustainability; circular economy; interpretive structural modelling; barriers; enablers; closed-loop supply chain; sustainable management

## 1. Introduction

Circular economy (CE) is increasingly colonizing industrial awareness [1,2] as a response to the pressure on Earth's life-support systems [3]. It requires a systemic transition, in which capabilities must be developed to slow and close material loops and thus decouple growth from the consumption of virgin materials. On both national and supranational levels, principles of a CE have been the basis of development strategies, e.g., in China's Circular Economy Promotion Law [4] and the European Union's Circular Economy Action Plan as a core building block of the European Green Deal [5]. Furthermore, adopting circular design strategies as well as activities of repair, reuse, remanufacture, or recycling are argued to yield competitive advantages [6] by improving environmental, financial, and social dimensions of a production system. Despite regulative targets and market benefits, manufacturing companies in particular struggle to transition towards higher degrees of circularity in a systemic way [7,8]. This is problematic due to their high resource consumption and waste pollution [9]. An extensive list of studies identifies barriers to such transition [10], many of which categorize barriers into several distinct clusters for communicative and systemization purposes. To exemplify, Ayati et al. [11] categorized barriers into eight groups, namely economics, governments and regulations, society and culture, technology, information and skills, markets, and organizations. Urbinati et al. [12] further introduced macro-level, meso-level, and micro-level barriers, and de Jesus and Mendonça [13] differentiated between hard barriers and soft barriers. Thus, several analytical

means of distinguishing barriers from one another exist, which generates an overview but tends to neglect the mutual dependencies among barriers. Kirchherr et al. [14] argued that the presence of barriers can both cause and be caused by a chain reaction of barriers. An example of a chain reaction is seen in low prices on virgin materials, which limits consumer demand and awareness. In turn, limited demand leads to a hesitant company culture, which further solidifies the linear lock-in [14,15]. Neglecting such interactions may cause practitioners to sub-optimize actions to promote a CE and ultimately lead to unsuccessful projects [14]. Consequently, companies must not only focus on identifying barriers but also assess how barriers are interlinked. Extant studies that explore this approach it through the lens of an interpretive structural modelling (ISM) methodology, as it is a well-established methodology in academia to examine linkages between variables, including barriers [16]. In the majority of these studies, the overview of dependencies among barriers acts as a basis for providing recommendations to managers and policy makers prescriptively. Little effort is made to thoroughly scrutinize the potential of the methodology to serve as a contextualization approach for practitioners to understand mutual dependencies among barriers as apparent to them in their respective context and approach them from a systemic perspective. The exact context-dependency of barriers [15] as well as approaching them from a systemic perspective [17] are often considered key elements to unlock the circular transition. This study addresses this research gap and is guided by the following research question:

> How can interpretive structural modelling be used as an approach to contextualize barrier interdependencies toward a circular economy?

The remainder of this paper is structured as follows. In Section 2, the theoretical background is delineated, which highlights the context dependency of barriers towards a CE as well as the application of an ISM methodology in a CE context. Section 3 presents the empirical foundation and the research methods. After this, an ISM is conducted in Section 4. Findings in terms of a case interpretation are presented in Section 5 and discussed in Section 6. Finally, the research objective is concluded in Section 7.

## 2. Theoretical Background

### 2.1. Context-Dependency of CE Barriers

Barriers towards a CE are multifaceted and appear on both employee, company, value chain, and institutional levels [18]. Nevertheless, the relative importance of individual barriers differs across companies depending on several contingencies [15]. To exemplify, de Jesus and Mendonça [13] showed that cultural barriers, e.g., inertia of business routines and acceptance of circular business models, and market-related barriers are the two least pressing groups of barriers. Opposed to this, Kirchherr et al. [14] showed that cultural barriers, e.g., "operating in a linear system" and "hesitant company culture", as well as market-related barriers are the two most pressing ones. As for technical know-how, too, extant literature presents opposing views. García-Quevedo et al. [19] argued that "lack of technical skills" constitutes a dominant challenge, whereas respondents from a survey by Ormazabal et al. [20] did not consider "lack of qualified personnel" as a challenge. Literature highlights company size as an important contingency factor. Small- and medium-sized enterprises (SMEs) often suffer from lower degrees of awareness and lack adequate technical capacity as well as resources to prioritize CE initiatives [21]. Large companies, on the other hand, might be better endowed to cope with high upfront investments [22]. Furthermore, supplier integration for sustainable initiatives is argued to be more challenging to SMEs due to limited bargaining power [21]. Finally, diverging industry-specific opportunities give rise to equally different barriers. For example, due to a well-established second-hand market, the automotive industry has long applied inner looping strategies, e.g., repair and reuse, to extend product lifetime. In recent years, automotive manufacturers have increasingly explored the potential of remanufacturing and recycling vehicles that are unsuitable for being repaired or reused. For this, non-destructive dismantling processes and quality concerns constitute major challenges, as they are tied to high operational costs

and uncertainty [23]. Harvesting material value through recycling has raised concerns, too, particularly with regard to the recyclability of electro-vehicle batteries [24]. The textile industry shares having a well-established second-hand market. However, due to only a few examples of textile remanufacturing or refurbishing, opportunities in the textile industry are often limited to repair, reuse, or recycle [25], which often takes place at a third-party retailer or recycling company. Thus, in contrast to the automotive industry, Jia et al. [26] showed that for textile manufacturers, technical barriers are of minor importance compared to market-related and organizational barriers. The abovementioned examples illustrate that barriers extensively differ depending on various contextual factors. Consequently, companies must identify barriers individually as perceived by their respective practitioners in their respective context.

*2.2. Interpretive Structural Modelling in a CE Context*

Despite an exponential increase in studies that adopt an ISM methodology [27], those that operate within a CE are scant. Sharma et al. [28] explored barriers to reverse logistics and argued that legal issues, financial constraints, and lack of awareness are key barriers to overcome, as they hold high driving power, whereas quality concerns and cooperation depend on other barriers. In the building sector, Bilal et al. [29] came to similar conclusions and introduced a mitigation framework to guide policy makers in accelerating a circular transition. In other studies, barriers are accompanied by drivers. Manoharan et al. [30] echoed the aforementioned findings while arguing that limited knowledge also causes a setback in implementation. As for drivers, the authors argued that stakeholder pressure holds high driving capabilities. In line with this, Patel et al. [31] showed that commitment from top management, globalization, and adequate environmental policies are key enablers that create fruitful conditions for other enablers, e.g., long-term strategic planning or training and education. Furthermore, a few studies identify links between industry 4.0 and CE. In an agricultural context, Kumar et al. [32] showed that governmental support, policies, and protocols must be in place to support practitioners in implementing industry 4.0 and CE practices. In the case of electric scooters, governmental support has proven equally important to the establishment of a sharing economy through investments in a suitable infrastructure [33]. Also tied to industry 4.0 as enabler of a CE, Rajput and Singh [34] highlighted process digitalization and interoperability as dominant barriers with high driving capabilities and argued that companies must focus on establishing a repository for information regarding post-market products. In line with this, Abdul-Hamid et al. [35] argued that technical and processual barriers are most dominant, e.g., lack of process design and difficulties in controlling a closed-loop supply chain. In contrast to other studies, however, they showed that financial constraints hold relatively high dependency power. Other studies adopted an ISM methodology to examine interlinkages among CE indicators [36] as well as factors that affect the selection of the most optimal third-party reverse logistics provider [37].

## 3. Empirical Foundation and Methods

Given the exploratory nature of this study, a single case study approach, as described by Yin [38], was adopted to illuminate mutual dependencies among the real-life barriers that industries encounter as they embark on a circular transition. In a CE context, large shares of extant literature are limited to identifying barriers and enablers either specific to industries or geographical settings. For this, surveys or multiple case studies, e.g., Mura et al. [39] or Gravagnuolo et al. [40], have proven valuable to identify tendencies across a broad sample of cases. However, in this study, understanding barriers and their mutual dependencies is not an end in itself. Rather, it is a means to an end to secure a knowledge base through which practitioners can obtain a contextualized understanding of the barriers that they encounter. Thus, the contribution of the study is not a generalizable overview of barriers to the circular transition but rather the provision of an industrially validated methodological approach to overcoming barriers based on their mutual depen-

dencies. Doing this requires in-depth contextual insights, for which a single case study is deemed useful. The empirical object of this study is a large Danish original equipment manufacturer (OEM) that produces mechatronic devices. Company characteristics are presented in Table 1. The mechatronics industry has been argued to face significant challenges in terms of adopting circular principles [41]. In 2020, the manufacturer engaged in a collaboration with Aalborg University to initiate its circular transition through the development of take-back systems. Therefore, case selection for this study builds upon the principle of convenience sampling, as described by Taherdoost [42]. More importantly, however, the case is considered relevant, as the company finds itself at an initial stage for which the identification of barriers as well as mitigation strategies currently receives significant attention. Consequently, practitioners exhibited enthusiasm and commitment to continuous involvement and testing.

**Table 1.** Company characteristics.

| Characteristics | |
|---|---|
| Size (employees) | 35,000+ |
| Industry | Mechatronics |
| Business model | Product-based |
| Customer segment | Industrial |
| Production strategy | Manufacture-to-order |
| Product customization | Functional customization |
| Product value | High |
| Product lifetime | 10–15 years |

The methodological contribution builds on two steps that are undertaken in close collaboration with the case company. These are described in the following.

### 3.1. Step 1: Identifying Barriers and Mutual Dependencies

The initial step contains a dual purpose. First, barriers to the circular transition were identified. To do this, 14 semi-structured interviews were conducted with practitioners from the selected case company. The limited sample size reflects the fairly low level of maturity of the case company in terms of undertaking a circular transition. This means that circular expertise resides among a small group of employees, all of whom were interviewed (see Table 2). However, to nuance such perspectives, it was considered important to not only include actors who are directly involved in CE implementation but to also provide a voice to peripheral actors with decision-making power, whose expertise is likely to affect the circular transition. Based on shared characteristics, barriers were then grouped and presented in a workshop, where interdependencies were discussed among participants. Second, to systemize the examination of mutual dependencies, the study adopted an ISM methodology. ISM is widely recognized as a valuable tool to systemize contextual relations among variables that comprise a specific challenge [43]. For complex topics, in particular, it has proven useful for visualizing causal effects, thus achieving a deeper understanding of the constituents of a problem, which can then be communicated to an audience [44]. Conducting an ISM follows a series of sub-steps, as argued by Attri et al. [16] and Abuzeinab et al. [45]. These were adapted to this specific study and are visualized in Figure 1. Furthermore, a detailed description of the sub-steps is provided along with the model development.

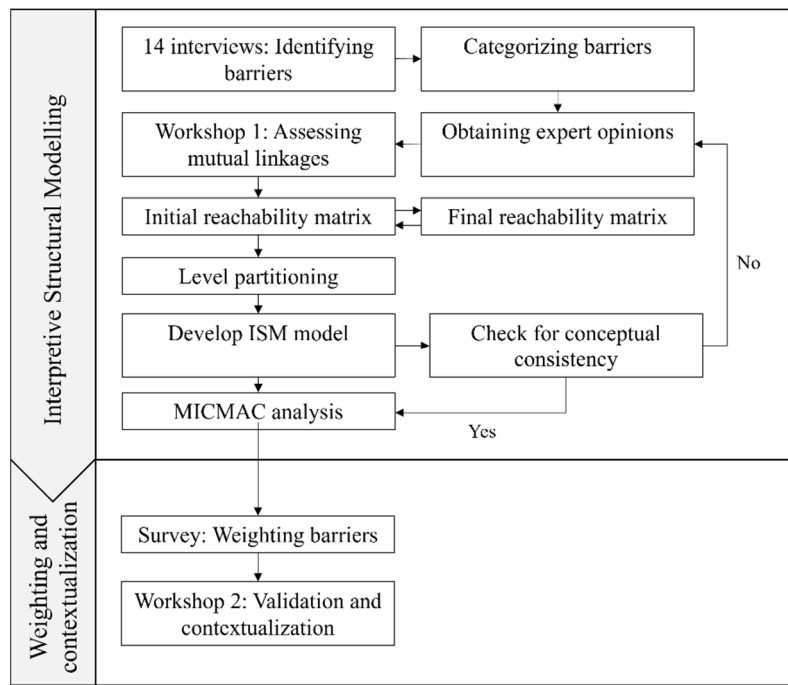

**Figure 1.** Research design of the study.

*3.2. Step 2: Weighting and Contextualization*

Having identified numerous barriers with high degrees of interrelatedness, it is considered important that companies are able to prioritize their effort to target barriers that are both possible to overcome or navigate around and are expected to have a high impact on the ability to implement a CE. It can be argued that an ISM methodology holds an embedded weighting, as targeting variables with high driving power creates fruitful conditions for addressing those with higher degrees of dependency power. An example of this is seen in Bilal et al. [29], who developed a mitigation framework that targets the variables with the highest driving power. A limitation to this is that the methodology itself puts little emphasis on the power dynamics of linkages between any given set of variables. To remedy this, the methodology is often supported by weighting measures, as evident in Kumar et al. [32]. In this study, interviewees and workshop participants were asked through a survey to weigh barriers on a Likert scale from 1–5 that determines the expected impact of overcoming a given barrier. For this, 1 refers to the least impactful barriers, while 5 refers to the most impactful barriers. The results were incorporated into the ISM model by color-coding the results from the survey; i.e., the darker the color, the more impactful the barrier is. Subsequent to this, a workshop was conducted with the same group of practitioners to validate the findings by stimulating reflections upon system dynamics.

**Table 2.** Overview of participants for data collection.

| Participants for Data Collection | | | | | |
|---|---|---|---|---|---|
| Title | Experience | Interview | Workshop 1 | Survey | Workshop 2 |
| Vice President | 15+ years | X | X | X | X |
| Senior Director (Engineering) | 15+ years | X | X | X | X |
| Standardization Manager | 0–5 years | X | X | X | X |
| People Leader | 15+ years | X | X | | |
| Head of Mechanics | 5–15 years | X | X | | X |
| Senior Director (Quality) | 15+ years | X | | | |
| Head of Global Sales | 15+ years | X | | | |

**Table 2.** *Cont.*

| Participants for Data Collection | | | | | |
|---|---|---|---|---|---|
| Title | Experience | Interview | Workshop 1 | Survey | Workshop 2 |
| Head of Sales (Marketing) | 5–15 years | X | | | |
| Engineering Director | 5–15 years | X | | X | |
| Process Excellence Manager | 5–15 years | X | | | |
| Senior Manager | 15+ years | X | X | | X |
| Project Manager | 5–15 years | X | X | X | X |
| Project Manager | 5–15 years | X | | X | X |

## 4. Model Development

### 4.1. Identifying Barriers

From the interviews, 14 barriers toward a circular transition were identified. These are clustered into five categories, i.e., managerial, market, financial, technical, and regulative. Table 3 provides an overview of this. Each barrier may contain multiple sub-barriers. One example of this is *complex reverse supply chain*, which covers the challenges of acquiring worn-out products as well as securing non-destructive disassembly. Consequently, the study acknowledges the plethoric barriers, as reported by Govindan and Hasanagic [10], but presents the ones identified by the interviewees on a higher level of detail to strike a balance between depicting barriers in a nuanced manner yet without compromising the interpretability of the ISM by having too many elements and mutual linkages.

**Table 3.** Identified CE barriers.

| Cluster | Barrier | Description |
|---|---|---|
| Managerial | Risk aversion (RA) | Managers are inclined to favor a complete overview of the circular transition. As this overview is often absent due to high uncertainties, managers are hesitant to take risks. |
| | Lack of internal coordination (LOC) | The ability to effectively undertake a circular transition requires a coordinative effort across functions, including but not limited to service centers, logistics, production, quality, and sales. This has proven particularly challenging. |
| | Lack of inspiration (LOI) | As CE is new to the case company, they are actively seeking inspiration from other companies. However, demonstration projects are scarce. |
| | Unclear visions (UV) | Circular economy has caught the awareness of the case company. Yet, visions for the transition are unclear. |
| | Lack of knowledge and competences (LKC) | Employees experience a lack of knowledge about the principles of a circular economy as well as the competences to integrate them into their daily operations. |
| Market | Lack of partnerships (LOP) | The case company acknowledges that a circular transition requires partnerships with customers, suppliers, third-party service partners, and/or waste handlers as well as universities in ways that differ from past collaborative efforts. However, little is known about the required capabilities from partners. |
| | Unclear sales strategy (USS) | Selling refurbished products is difficult due to fluctuating availability. As the product return flow is unstable, availability of products cannot be guaranteed. Furthermore, questions are raised concerning sales channels. |
| | Lack of customer demand and acceptance (LDA) | Customer demand remains weak. Furthermore, it is questionable as to what degree customers are willing to accept changes, e.g., for the visual appearance of refurbished products. |

**Table 3.** *Cont.*

| Cluster | Barrier | Description |
|---|---|---|
| Financial | Poor profitability (PP) | It is difficult for a take-back program to generate a profitable business case in the short term. Long-term profitability is considered probable but with high uncertainty. |
| Technical | Complex reverse supply chain (RSC) | Complexity of developing a reverse supply chain is high due to difficulties of acquiring products as well as product disassembly. |
| Technical | Lack of circular design (LCD) | As products on the market have not been designed for a circular economy, disassembly of products is significantly hampered. |
| Technical | Questionable reliability (QR) | As the case company produces high-quality products, concerns are raised about the reliability of refurbished products. |
| Regulative | Obstructing regulation (OR) | Obstructing regulation hampers take-back. To exemplify, end-of-life products are sometimes considered waste, which makes it difficult to import/export across borders. |
| Regulative | Lack of incentives (LI) | Few incentives are provided by national or international regulations. |

*4.2. Structural Self-Interaction Matrix*

After the identification of barriers, a structural self-interaction matrix (SSIM) was developed (see Table 4). The purpose is to explore contextual relationships among the barriers. According to Attri et al. [16], this step should be conducted in collaboration with experts who are familiar with the object of inquiry. For this study, the group of experts was comprised of industrial practitioners to generate a model that emerges from locally grounded knowledge. A workshop was conducted with eight participants. After being introduced to the barriers, people were divided into two groups to assess contextual relations based on whether a given barrier influences another one. These relationships were assessed following the logic in which i refers to rows, while j refers to columns:

V: Barrier i influences barrier j;
A: Barrier j influences barrier i;
X: Both barriers affect each other;
O: No relationship.

**Table 4.** Structural self-interaction matrix.

|  | LI | OR | QR | LCD | RSC | PP | LDA | USS | LOP | LKC | UV | LOI | LOC | RA |
|---|---|---|---|---|---|---|---|---|---|---|---|---|---|---|
| RA | A | A | X | O | X | X | X | X | V | X | X | X | A | |
| LOC | O | O | O | V | X | X | A | V | X | V | A | A | | |
| LOI | O | O | V | V | V | V | A | V | X | X | X | | | |
| UV | A | O | O | V | V | X | A | X | V | X | | | | |
| LKC | A | A | V | V | V | X | X | X | X | | | | | |
| LOP | X | A | A | X | X | X | X | X | | | | | | |
| USS | X | A | A | X | X | X | X | | | | | | | |
| LDA | V | A | X | X | X | X | | | | | | | | |
| PP | X | A | X | X | X | | | | | | | | | |
| RSC | A | A | O | A | | | | | | | | | | |
| LCD | A | A | X | | | | | | | | | | | |
| QR | A | A | | | | | | | | | | | | |
| OR | X | | | | | | | | | | | | | |
| LI | | | | | | | | | | | | | | |

### 4.3. Reachability Matrix

Based on the SSIM, an initial reachability matrix was developed, which is a binary matrix that provides an overview of the driving power and dependency power of each barrier. The conversion follows the logic as argued by Attri et al. [16]:

- If the (i,j) entry in the SSIM is V, then the (i,j) entry in the reachability matrix becomes 1, and the (j,i) entry becomes 0.
- If the (i,j) entry in the SSIM is A, then the (i,j) entry in the reachability matrix becomes 0, and the (j,i) entry becomes 1.
- If the (i,j) entry in the SSIM is X, then the (i,j) entry in the reachability matrix becomes 1, and the (j,i) entry becomes 1.
- If the (i,j) entry in the SSIM is O, then the (i,j) entry in the reachability matrix becomes 0, and the (j,i) entry becomes 0.

The initial reachability matrix (Table 5) reveals a plethora of direct linkages among barriers. This means that as it undergoes a transitivity check that accounts for multiple intermediate linkages to develop a final reachability matrix (Table 6), every barrier affects one another. This clearly illustrates the complex interrelations as perceived by practitioners that companies face as they embark on a circular transition, which is often tied to high degrees of uncertainty. Such interrelatedness may arguably be a result of the industrial embeddedness, i.e., asking practitioners to both identify barriers and assess their mutual dependencies. Nevertheless, if such interrelations are to be interpreted and utilized by practitioners, it provides little guidance for them if all barriers hold similar dependency power and driving power. Therefore, while acknowledging the importance of transitivity, the authors suggest for this case to use the initial reachability matrix as outset for level partitioning, as it generates a nuanced multi-level model, based on which practitioners can prioritize their effort. This moderation builds upon the notion of a consensus-shifting rather than a consensus-creating theoretical contribution [46], as it brings nuances to the use of transitivity. Although transitivity has proven valuable in extant studies of complex phenomena, including the circular transition, the phenomena and its variables may occasionally become too entangled with a plethora of linkages, which may hamper its purpose to assist individuals or groups in understanding the structure of a complex issue [16].

**Table 5.** Initial reachability matrix.

|  | LI | OR | QR | LCD | RSC | PP | LDA | USS | LOP | LKC | UV | LOI | LOC | RA | Driv. Power |
|---|---|---|---|---|---|---|---|---|---|---|---|---|---|---|---|
| RA | 0 | 0 | 1 | 0 | 1 | 1 | 1 | 1 | 1 | 1 | 1 | 1 | 0 | 1 | 10 |
| LOC | 0 | 0 | 0 | 1 | 1 | 1 | 0 | 1 | 1 | 1 | 0 | 0 | 1 | 1 | 8 |
| LOI | 0 | 0 | 1 | 1 | 1 | 1 | 0 | 1 | 1 | 1 | 1 | 1 | 1 | 1 | 11 |
| UV | 0 | 0 | 0 | 1 | 1 | 1 | 0 | 1 | 1 | 1 | 1 | 1 | 1 | 1 | 10 |
| LKC | 0 | 0 | 1 | 1 | 1 | 1 | 1 | 1 | 1 | 1 | 1 | 1 | 0 | 1 | 11 |
| LOP | 1 | 0 | 0 | 1 | 1 | 1 | 1 | 1 | 1 | 1 | 0 | 1 | 1 | 0 | 10 |
| USS | 1 | 0 | 0 | 1 | 1 | 1 | 1 | 1 | 1 | 1 | 1 | 0 | 0 | 1 | 10 |
| LDA | 1 | 0 | 1 | 1 | 1 | 1 | 1 | 1 | 1 | 1 | 1 | 1 | 1 | 1 | 13 |
| PP | 1 | 0 | 1 | 1 | 1 | 1 | 1 | 1 | 1 | 1 | 1 | 0 | 1 | 1 | 12 |
| RSC | 0 | 0 | 0 | 0 | 1 | 1 | 1 | 1 | 1 | 0 | 0 | 0 | 1 | 1 | 7 |
| LCD | 0 | 0 | 1 | 1 | 1 | 1 | 1 | 1 | 1 | 0 | 0 | 0 | 0 | 0 | 7 |
| QR | 0 | 0 | 1 | 1 | 0 | 1 | 1 | 1 | 1 | 0 | 0 | 0 | 0 | 1 | 7 |
| OR | 1 | 1 | 1 | 1 | 1 | 1 | 1 | 1 | 1 | 1 | 0 | 0 | 0 | 1 | 11 |
| LI | 1 | 1 | 1 | 1 | 1 | 1 | 0 | 1 | 1 | 1 | 1 | 0 | 0 | 1 | 11 |
| Depend. Power | 6 | 2 | 9 | 12 | 13 | 14 | 10 | 14 | 14 | 11 | 8 | 6 | 7 | 12 |  |

**Table 6.** Final reachability matrix, in which * refers to transitive relations.

|  | LI | OR | QR | LCD | RSC | PP | LDA | USS | LOP | LKC | UV | LOI | LOC | RA | Driv. Power |
|---|---|---|---|---|---|---|---|---|---|---|---|---|---|---|---|
| RA | 1 * | 1 * | 1 | 1 * | 1 | 1 | 1 | 1 | 1 | 1 | 1 | 1 | 1 * | 1 | 14 |
| LOC | 1 * | 1 * | 1 * | 1 | 1 | 1 | 1 * | 1 | 1 | 1 | 1 * | 1 * | 1 | 1 | 14 |
| LOI | 1 * | 1 * | 1 | 1 | 1 | 1 | 1 * | 1 | 1 | 1 | 1 | 1 | 1 | 1 | 14 |
| UV | 1 * | 1 * | 1 * | 1 | 1 | 1 | 1 * | 1 | 1 | 1 | 1 | 1 | 1 | 1 | 14 |
| LKC | 1 * | 1 * | 1 | 1 | 1 | 1 | 1 | 1 | 1 | 1 | 1 | 1 | 1 * | 1 | 14 |
| LOP | 1 | 1 * | 1 * | 1 | 1 | 1 | 1 | 1 | 1 | 1 | 1 * | 1 | 1 | 1 * | 14 |
| USS | 1 | 1 * | 1 * | 1 | 1 | 1 | 1 | 1 | 1 | 1 | 1 | 1 * | 1 * | 1 | 14 |
| LDA | 1 | 1 * | 1 | 1 | 1 | 1 | 1 | 1 | 1 | 1 | 1 | 1 | 1 | 1 | 14 |
| PP | 1 | 1 * | 1 | 1 | 1 | 1 | 1 | 1 | 1 | 1 | 1 | 1 * | 1 | 1 | 14 |
| RSC | 1 * | 1 * | 1 * | 1 * | 1 | 1 | 1 | 1 | 1 | 1 * | 1 * | 1 * | 1 | 1 | 14 |
| LCD | 1 * | 1 * | 1 | 1 | 1 | 1 | 1 | 1 | 1 | 1 * | 1 * | 1 * | 1 * | 1 * | 14 |
| QR | 1 * | 1 * | 1 | 1 | 1 * | 1 | 1 | 1 | 1 | 1 * | 1 * | 1 * | 1 * | 1 | 14 |
| OR | 1 | 1 | 1 | 1 | 1 | 1 | 1 | 1 | 1 | 1 | 1 * | 1 * | 1 * | 1 | 14 |
| LI | 1 | 1 | 1 | 1 | 1 | 1 | 1 * | 1 | 1 | 1 | 1 | 1 * | 1 * | 1 | 14 |
| Depend. Power | 14 | 14 | 14 | 14 | 14 | 14 | 14 | 14 | 14 | 14 | 14 | 14 | 14 | 14 |  |

### 4.4. Partitioning into Levels

Based on the initial reachability matrix, level partitions can be instigated (see Table 7). This is carried out by deriving reachability and antecedent sets from the matrix. A reachability set refers to the barriers that a given barrier affects, including itself. On the contrary, an antecedent set refers to the barriers that a given barrier is affected by, including itself. Intersection presents the barriers that the reachability set and the antecedent set share. Partitioning this into levels is carried out in iterations. First, if the reachability set and the intersection are identical for any given barriers, they enter the model as top-level barriers. For the subsequent iterations, these barriers are disregarded. This process is repeated for the following levels. From this, a digraph is constructed, in which levels as well as linkages are depicted [16]. This generates the final ISM model, as presented in Figure 2.

**Table 7.** Level partitions based on reachability set, antecedent set, and intersections.

| Barriers | Reachability | Antecedent | Intersection | Level |
|---|---|---|---|---|
| RA | RA,QR,RSC,PP,LDA, USS,LOP,LKC,UV,LOI | RA,LOC,LOI,UV,LKC,USS, LDA,PP,RSC,QR,OR,LI | RA,QR,RSC,PP,LDA, USS,LKC,UV,LOI | II |
| LOC | LOC,LCD,RSC,PP,USS, LOP,LKC,RA | LOC,LOI,UV,LOP,LDA,PP, RSC | LOC,RSC,PP,LOP | VI |
| LOI | LOI,QR,LCD,RSC,PP, USS,LOP,LKC,UV, LOC,RA | LOI,RA,UV,LKC,LDA | LOI,LKC,UV,RA | VI |
| UV | UV,LCD,RSC,PP,USS, LOP,LKC,LOI,LOC,RA | UV,RA,LOI,LKC,USS,LDA, PP,LI | UV,PP,USS,LKC,UV, LOI,RA | IV |
| LKC | LKC,QR,LCD,RSC,PP, LDA,USS,LOP,UV,LOI, RA | LKC,RA,LOC,LOI,UV,LOP, USS,LDA,PP,OR,LI | LKC,PP,USS,LOP,UV, LOI,RA | V |
| LOP | LOP,LI,LCD,RSC,PP, LDA,USS,LKC,LOI,LOC, RA | LOP,RA,LOC,LOI,UV,LKC, USS,LDA,PP,RSC,LCD,QR, OR,LI | LOP,LI,LCD,RSC,PP, LDA,USS,LKC,LOI, LOC,RA | I |

**Table 7.** *Cont.*

| Barriers | Reachability | Antecedent | Intersection | Level |
|---|---|---|---|---|
| USS | USS,LI,LCD,RSC,PP, LDA,LOP,LKC,UV,RA | USS,RA,LOC,LOI,UV,LKC, LOP,LDA,PP,RSC,LCD,QR, OR,LI | USS,LI,LCD,RSC,PP, LDA,LOP,LKC,UV,RA | I |
| LDA | LDA,LI,QR,LCD,RSC,PP, USS,LOP,LKC,UV,LOI, LOC,RA | LDA,RA,LKC,LOP,USS,PP, RSC,LCD,QR,OR,LI | LDA,LI,QR,LCD,RSC, PP,USS,LOP,LKC,RA | VIII |
| PP | PP,LI,QR,LCD,RSC,LDA, USS,LOP,LKC,UV,LOC, RA | PP,RA,LOC,LOI,UV,LKC, LOP,USS,LDA,RSC,LCD, QR,OR,LI | PP,LI,QR,LCD,RSC, LDA,USS,LOP,LKC,UV, LOC,RA | II |
| RSC | RSC,PP,LDA,USS,LOP, LOC,RA | RSC,RA,LOC,LOI,UV,LKC, LOP,USS,LDA,PP,LCD,OR, LI | RSC,PP,LDA,USS,LOP, LOC,RA | I |
| LCD | LCD,QR,RSC,PP,LDA, USS,LOP | LCD,LOC,LOI,UV,LKC, LOP,USS,LDA,QR,OR,LI | LCD,QR,LDA,USS,LOP | III |
| QR | QR,LCD,PP,LDA,USS, LOP,RA | QR,RA,LOI,LKC,LDA,PP, LCD,OR,LI | QR,LCD,PP,LDA,RA | II |
| OR | OR,LI,QR,LCD,RSC,PP, LDA,USS,LOP,LKC,RA, | OR,LI | OR,LI | IX |
| LI | LI,OR,QR,LCD,RSC,PP, USS,LOP,LKC,UV,RA | LI,LOP,UDD,LFS,PP,OR | LI,OR,PP,USS,LOP | VII |

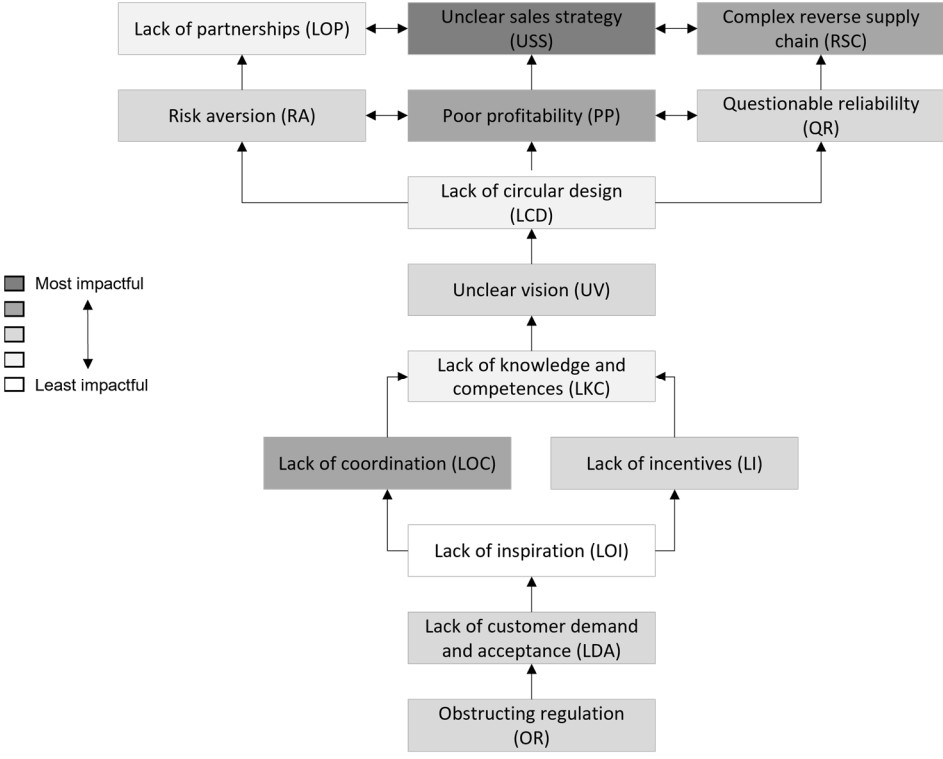

**Figure 2.** Weighted ISM model.

## 4.5. MICMAC Analysis

Subsequent to the level partitioning, a MICMAC (cross-impact matrix multiplication applied to classification) analysis was conducted (see Figure 3). For this, barriers were categorized into four clusters based on their driving power and dependency power. Barriers with low driving power and low dependency power are referred to as autonomous, as

they are weakly tied to the studied system. Barriers with high driving power and low dependency power are referred to as independent barriers. Barriers with low driving power and high dependency power are referred to as dependent barriers, while those with high driving power and high dependency power are referred to as linkage barriers [16].

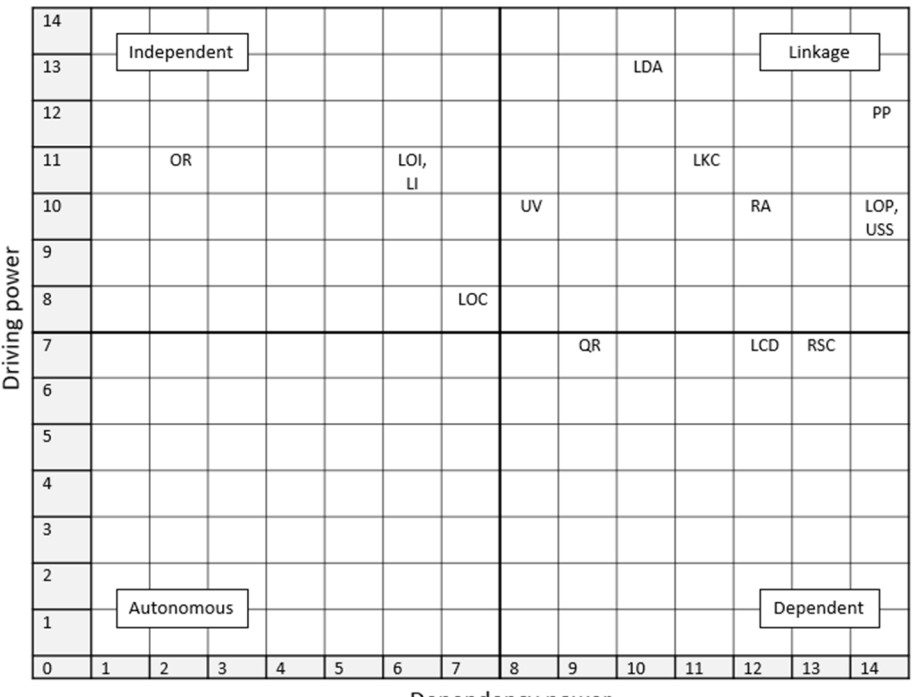

**Figure 3.** Categorizing barriers into clusters in a MICMAC analysis.

## 5. Results

This section explains and exemplifies how ISM has proven to support practitioners in obtaining an understanding of the contextual embeddedness of CE barriers. It is widely agreed that the circular transition requires a systemic approach to account for the mutual dependencies of all elements of an organizational system [47]. Without this, companies risk sub-optimizing by refraining from detecting potential root causes. The need for a systemic approach is supported by Figure 3, in which it becomes evident that CE barriers are embedded within a complex interplay of mutual dependencies due to the abundance of linkage barriers and no autonomous barriers. Such linkage barriers are difficult to approach, as any attempt to tackle them affects the dynamics among other barriers, which in turn affects the linkage barriers themselves. From the weighting, it appears that *obstructing regulation* and *lacking customer demand and acceptance*, although they both hold high driving power, are not expected to be the most impactful barriers. Thus, as antecedents of other barriers, their linkages are relatively weak. The most impactful barriers are fairly dispersed. *Unclear sales strategy* appears to be the most impactful and thus critical barrier to address. Two other barriers that possess high dependency power and are expected to have a high impact are *poor profitability* and *complex reverse supply chain*. Last, *lack of coordination* is considered of the same criticality. These are the four most critical factors as perceived by practitioners from the case company, and overcoming these are expected to significantly enable a circular transition. Yet, unlocking these critical barriers requires a scrutinization of the network of other barriers in which they are situated.

It becomes evident that having unclear sales strategies holds a particularly high dependency power. For the case company, the criticality is tied to the uncertainty about product specifications, which causes the sales organization to refrain from actively scouting for available sustainable alternatives. Such challenge, however, becomes solidified in

the organization due to the presence of other barriers. The reverse supply chain being complex has proven to create difficult conditions for the sales organization to determine a "road-to-market" strategy. A core cause for this is tied to the fluctuating availability of circulated devices, particularly in the early phases of the circular transition, where volume is low, and companies are developing and raising awareness about take-back capabilities. For the case company, this becomes even more difficult, as it sells products to an industrial market with the vast majority of sold products being customized. Both sales strategies and the reverse supply chain are further challenged by the lack of partnerships. The company finds it difficult to identify relevant partners and engage in collaborative efforts, e.g., with strategically important and innovative customers. Engaging in such supplier–buyer partnerships could entail a closed-loop supply chain, in which customers agree to return end-of-life devices to the manufacturer as well as buy circulated devices. Such agreement would bring about a degree of flexibility for the manufacturer to conduct niche experiments and test assumptions for the reverse logistics setup as well as pricing mechanisms. Supply chain collaboration is often highlighted as a core lever for a circular transition, but it remains limited due to various constraints. This study finds that risk aversion from managers hampers the willingness to seize collaborative opportunities. Such hesitation is tied to plethoric uncertainties. Opposed to what one might think, concerns were not related to challenges of finding appropriate partners. Rather paradoxically, concerns revolved around the scenario in which market interest for sustainable alternatives would increase to a degree, where the manufacturer would be unable to meet the market demands and thus jeopardize customer relations. This would not be an issue but a great opportunity were it not for the two additional uncertainties, i.e., questionable reliability and poor profitability. In terms of reliability, managers are not confident in dealing with questions of quality. Should a remanufactured product be as good as new and be able to pass existing quality control, or is it acceptable to offer it as an alternative with a lower environmental footprint, but it comes with a shorter expected lifetime? This is particularly relevant for the case company, which produces durable products with a long lifetime expectancy, as many last for 10–15 years (see Table 1).

Underpinning the complex interrelatedness, such considerations are core determinants for being able to define a sales strategy. As for profitability, managers raise concerns about potentially being locked in a path that has proven hardly profitable. Several factors feed into such concerns as core dependencies, including high costs of establishing a reverse supply chain with product acquisition mechanisms as well as manual inspection and non-destructive disassembly processes. The notion of profitability, as exhibited by managers, refers to a revenue generation that exceeds operational costs. Although sustainable initiatives may yield competitive advantage and increase market demand for other product portfolios, it remains difficult to account for this in financial feasibility studies. High operational costs, as inhibiting profitability, are extensively challenged by the lack of circular product design. As described by managers, decommissioned products, regardless of whether they are new to market or legacy products, have not been designed for being easily disassembled to repair or replace worn-out components. To provide a case-specific example, this means that mechanical components and electronic components are often glued together, which inhibits recovery opportunities. Being able to separate such components would be beneficial, as their expected lifetime as components significantly differs. To a great extent, the lack of circular design is rooted in barriers that tie to governance mechanisms for the circular transition. Product designers have refrained from actively considering design strategies, such as "design-for-disassembly" or increased modularization, due to unclear visions. Therefore, they have not had a clear managerial mandate for integrating design principles that favors circularity, nor have they been guided by circular performance measures. The challenge of unifying a direction for the company as well as making it pervade into various departments highly affects the perceived lack of coordination, which suffers from the absence of a materialized long-term strategy. Although practitioners acknowledge the importance of a cross-functional team, little is known about the required expertise from

other functions to reach a rather undefined scenario for a circular future, nor does it aid managers in assessing needs for additional knowledge and competences in terms of talent acquisition. Difficulties in developing appropriate governance structures, including visions and coordination, are shown to be solidified by the lack of inspiration. Due to limited experience, managers are searching for inspiration, but sources of inspiration are scarce. This is extensively caused by limited market demand for remanufactured products. As experienced by managers, it remains questionable as to what degree customers are willing to accept changes in product and material composition, e.g., by including recycled plastics or remanufactured mechanical components. Following an increase in market demand, managers would be further incentivized in allocating resources to govern the circular transition and conduct demonstration projects to test viability of new market opportunities. However, limited market demand remains rooted in regulation that hardly favors a more regenerative economy through incentives, such as taxation on virgin materials, but in some cases obstruct the implementation of take-back activities. This is shown to hold the highest driving power relative to its dependency power, and ultimately, overcoming such barrier is expected to create fruitful conditions for overcoming other barriers with higher degrees of dependency.

## 6. Discussion

The purpose of the study has been to explore the use of ISM as an approach to assist practitioners in systematically obtaining a contextualized understanding of CE barriers and their mutual dependencies. This has proven necessary, as barriers are argued to be highly context-dependent while being embedded within a complex interplay of relationships. Consequently, this research has been guided by the following research question: How can interpretive structural modelling be used as an approach to contextualize barrier interdependencies towards a circular economy?

The core contribution ties to the notion of contextualization through ongoing involvement of practitioners throughout the process. This is in contrast to extant literature, e.g., Manoharan et al. [30] and Kumar et al. [32], which seeks to provide a generalizable overview of barriers and drivers within the automobile and the agricultural industry, respectively, by involving experts from industry and research to discuss barriers, as identified in literature. Although this is valuable for understanding the apparent issues for the circular transition at an industry level, several contingencies influence the relative weight of each individual barrier at a company level. Adopting generic barriers as a backdrop for developing mitigation strategies might cause practitioners to target disproportionate root causes and construct a misguiding path of mitigation measures to unlock critical barriers. To exemplify the context dependency, this study finds that regulation highly influences other barriers. This coheres with the findings from Kumar et al. [32] and Sharma et al. [28] but differs from those of Ravi and Shankar [48] and Makki et al. [49]. Nevertheless, the needs for regulation might extensively differ. If companies are engaged in product take-back activities, challenges might become apparent in the shape of obstructing regulation, e.g., in cases where import/export regulation hampers transportation of worn-out devices across borders to remanufacturing sites, as they are often being characterized as waste [14]. Other companies might be more inclined towards pushing for regulation and standards to supply the framework conditions for an effective market, e.g., through taxation instruments or requirements for product and material composition. Such different orientations create equally different conditions for overcoming barriers with higher dependency. The composition of barriers, including their mutual dependencies, is furthermore likely to mirror the level of maturity for the circular transition as perceived by the respective company. Companies, including the case of this study, that find themselves in early levels of the circular transition, e.g., what Uhrenholt et al. [50] referred to as "basic" or "explorative" levels, may delegate high degrees of driving power to elements of governance, including barriers of outlining visions, coordinating the transition, as well as prioritizing the acquisition of new knowledge and competences. Companies with higher levels of maturity, on the other hand, may be more

oriented towards formalizing and standardizing procedures to fully integrate CE as an organizational priority.

### 6.1. Managerial Implications

This study builds upon the argumentation that managers should refrain from approaching CE barriers as being independent of each other, as they risk sub-optimizing their efforts. Instead, they must scrutinize the complex interplay of mutual dependencies as identified and interpreted by themselves in their respective context. This study proposes ISM as an approach for obtaining such understanding. It can be adopted by practitioners as an enabler of contextualization to support them in strategic planning as well as project-planning activities. However, as experienced in this process, it is important to have practitioners aligned in terms of scope to prevent a situation where some are anchored in project-specific barriers, while others are viewing CE in the light of a long-term organizational transformation. Although there might be several similarities, the relative importance of barriers might also differ due to varying time horizons. Thus, the scope of the process must be clarified at the outset. Related to this, the approach can preferably be used ex ante to generate a weighted overview of barriers prior to the allocation of resources. However, as experienced in this study, it has also proven valuable as an ex-post approach to explain why the company was not progressing at the expected pace. For example, it had proven difficult for the case company to scope, plan, and carry out rapid demonstration projects through a bottom-up approach. By focusing on mutual dependencies among barriers, practitioners found that such challenges could be explained by the numerous dependencies of operational barriers. Therefore, they figured that they must simultaneously develop governance mechanisms to legitimize product take-back initiatives and support them through investments and strategic planning. Besides this, managers are recommended to involve a cross-functional team throughout the process, from identifying barriers to contextualizing their mutual dependencies. This is expected to generate more diversified perspectives to qualify and nuance discussions while serving as a vehicle for increased awareness across functions. Ultimately, a contextualized understanding of dependencies is expected to enable managers to outline mitigation strategies. As evident from this study as well as extant literature, governmental support holds high driving power relative to its dependency power. Although higher degrees of regulative support are expected to create fruitful conditions for overcoming other barriers, it might not be the ideal starting point for all companies, as it is likely to be a slow process, which conflicts with the urgent need for climate action. Consequently, managers must scrutinize the mutual dependencies to identify actions and mitigation strategies that are meaningful to them.

### 6.2. Academic Contribution

Extant literature has extensively explored CE barriers but fails to identify mutual dependencies. This is despite evidence that barriers are highly context-dependent and that they are embedded within a complex network. Therefore, they cannot be targeted individually. The academic contribution of this study ties to the introduction of a methodological approach that builds upon an established methodology but explores its use as an approach to contextualize barriers rather than point towards generalizability. This has been tested in an industrial setting based on which managerial recommendations are provided. Thus, the novelty lies within the reframing of barriers as being mutually dependent as well as the proposition of an approach to systemize such interrelation to be centered around the mutual dependencies rather than trying to overcome barriers independent from one another. Furthermore, from this process, the study aids in concretizing examples of chain mechanisms among barriers, as scarcely argued in extant literature.

## 7. Conclusions and Limitations

In the context of a circular transition, this study proposes an ISM-based methodological contribution to enable companies in contextualizing mutual dependencies among barriers.

This has proven valuable, as the presence of several chain mechanisms among barriers, combined with them being dependent on several contingencies, calls for an approach that guides practitioners in structuring such process while tailoring it to their respective context. This approach is divided into two phases. First, an ISM methodology was used to examine mutual dependencies among barriers, as identified through a series of interviews with practitioners. For this study, 14 barriers were identified and connected to each other to illustrate their respective driving and dependency power. Second, barriers were weighted on a Likert scale from 1–5 to identify critical barriers. This revealed four critical barriers, i.e., *unclear sales strategy*, *poor profitability*, *complex reverse supply chain*, and *lack of coordination*. Unlocking these barriers requires a thorough understanding of their embeddedness in the network. Through a workshop, practitioners scrutinized the mutual dependencies to obtain such understanding and proved capable of using it as a visual backdrop for discussions to contextualize barriers and trace back dependencies to identify root causes that inhibit them from instigating and accelerating the circular transition.

The study holds several limitations. As part of the first workshop, participants were divided into two groups, each of which discussed relationships between half of the barriers. This caused participants to adopt various levels of understanding as to how direct barriers should tie to one another for a linkage to be defined. Therefore, one group identified significantly more two-way relationships than the other group. Due to this, the ISM (Figure 2) was based on the initial reachability matrix rather than the final reachability matrix. This meant that the study primarily focuses on the direct relationships and not the indirect ones, which could have strengthened the cognitive network formation. Another limitation relates to the adoption of a single case study methodology. While it is argued that such methodology aids in providing in-depth understanding of a complex phenomenon throughout different phases of the study, the absence of multiple cases causes the validity of the methodological contribution to be questionable. Future research is encouraged to replicate the propositions of this study in multiple cases, preferably in different contexts, to both illustrate the contingency dependence of barriers and diversify the findings of this study.

**Author Contributions:** S.F.J., conceptualization, methodology, formal analysis, writing—original draft, and writing—review and editing; J.H.K., conceptualization, writing—review and editing, and supervision; J.N.U., conceptualization and writing—review and editing; M.C.R., conceptualization and writing—review and editing; S.A., conceptualization and writing—review and editing; B.V.W., conceptualization, writing—review and editing, and supervision. All authors have read and agreed to the published version of the manuscript.

**Funding:** This research has been funded by the Manufacturing Academy of Denmark (MADE) as part of the innovation program MADE Fast. Funding nr.: 9090-00002B.

**Informed Consent Statement:** Informed consent was obtained from all subjects involved in the study.

**Data Availability Statement:** Data are available upon request.

**Conflicts of Interest:** The authors declare no conflict of interest.

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
