# Peer review of "Unlocking Barriers to Circular Economy: An ISM-Based Approach to Contextualizing Dependencies"

_sustainability, doi:10.3390/su14159523_

Round 1
Reviewer 1 Report
Dear Authors,
You paper on the circular economy is well written, is easy to understand, and is a relevant contribution to the current discussions on the subject.
Reviewer 2 Report
The paper reports investigation on barriers in circular economy practices. This research is interesting, and the results were appropriately presented. Overall, the paper was well written. However, the paper can be further improved as the following:
(1) The literature analysis in the Introduction section is scattered and needs to be streamlined and concentrated. The research gaps and why the ISM model is adopted should be explained.
(2) Figure 1 shows 14 semi-structured interviews have been conducted. It is a small sample. The authors should justify the findings that are constructed on those interviews.
(3) CE often involves complex interrelations in organizations. However, this study only identify 14 barriers (p5 line 187) for further analysis. It may be reasonable for SME, but not for large enterprise or international organizations.
(4) The analysis in “4. Results” are performed based on a case study. The findings failed to be linked with the product/service as well as the operations of the company.
Reviewer 3 Report
I read with interest the manuscript titled “Unlocking barriers to circular economy: An ISM-based approach to contextualizing dependencies.” I think it is an interesting and important topic. However, I have some comments to be addressed as follows:
Abstract:
The thesis statement in lines 13 and 14 is very general and not specific to the main topic, "circular economy." I suggest rewriting the sentence.
There is no information presented on data collection and participants in the stud in the abstract. Please add a sentence.
The findings of the study in lines 16 and 17 are very general. It is expected that the application of ISM will result in a network of interconnected barriers. Therefore, please include the study's key findings in the context of a "circular economy."
Implications of the study in lines 17-20, please be more specific on how the developed model can help and who? Which "practitioners"? And how do they work "as levers for the circular transition"?
Keywords:
The words "barriers,"; "circular economy,"; "ISM," which are present in the title, were not included.
Methodology:
According to Figure 1 and lines 151 and 152, the literature review is input to identifying the barriers. So, please cite each of the barriers in Table 2 with associated references from where they were extracted.
As indicated in lines 153 and 154, 14 interviews were conducted with practitioners. Does this also mean that 14 practitioners were engaged in the study? Please add a table of practitioner profiles, including their specialty, years of experience, role, etc. related to the circular economy.
The ISM is well known for its seven steps. The presented analysis lacks some:
1. A final reachability matrix (FRM). This ISM step is very important to check for any transitive relationships between the studied barriers. The argument in lines 224-227 is weak. The whole point of using ISM is to represent the complex cognitive network of direct and indirect interrelationships between barriers in participants' minds based on their experience. And basing the model on the initial reachability matrix (IRM) alone means considering direct relations only and ignoring higher order indirect ones. Please apply a transitivity test on (IRM) presented in Table 4 and produce and present an FRM. After doing so, it is expected that the driving and dependence powers of barriers will change. Thus go the 2nd point.
2. Repeat the analysis of the partitioning matrix (PM) iterations according to the new FRM.
3. MICMAC analysis is missing. It is an important ISM step as it classifies the barriers into four categories: 1. autonomous, 2. linkage, 3. dependent, and 4. independent barriers in a quadrant chart. This will provide insights on the barriers and their roles in the studied problem and advise the construction of the ISM network diagraph.
4. If the above 3 points are met, it is expected that a new network might result. Therefore, add the missing steps to the methodology and modify the analysis, discussion, and conclusions accordingly.
"Weighting and contextualization" is an interesting idea. In line 180, it is indicated that it was done using a 5-point Likert scale. In line 181, it is stated that "The results have been incorporated into the ISM model." How? Please elaborate more on the used scale and on this incorporation. Also, how used to color code the barriers in Figure 2.
Please consider the following references:
Ahmad, N.; Qahmash, A. SmartISM: Implementation and Assessment of Interpretive Structural Modeling. Sustainability 2021, 13, 8801. https://doi.org/10.3390/su13168801
Makki, A.A.; Alidrisi, H.; Iqbal, A.; Al-Sasi, B.O. Barriers to green entrepreneurship: an ISM-based investigation. Journal of Risk and Financial Management 2020, 13(11), 249. https://doi.org/10.3390/jrfm13110249
Warshall, S. A theorem on boolean matrices. Journal of the ACM (JACM) 1962, 9(1), 11-12.
Mandal, A.; Deshmukh, S. Vendor selection using interpretive structural modelling (ISM). International journal of operations & production management 1994.
Please, proofread the manuscript.
Round 2
Reviewer 2 Report
I am happy with the revisions and I suggest to accept the paper.
Author Response
Once again, thank you for taking your time to review the manuscript.

Reviewer 3 Report
I would like to thank the authors for considering my comments and suggested modifications. I believe the paper is better now.
Only two minor points need to be addressed:
1. Please check the final reachability matrix (FRM) in Table 6. Check the barrier (OR) column. I believe more transitive relationships exist with this barrier based on the initial reachability matrix (IRM) in Table 5. I think all driving, and dependence power sums in the FRM will all add up to 14 in each row and each column based on Warshall's algorithm. This will strengthen the argument of basing the model of the IRM.
2. The MICMAC analysis should be presented as a figure, not a table, because it is a chart. Therefore, please modify the caption of Table 8 to be a caption of Figure 3 and move it under the MICMAC chart.
Good luck!
Author Response
(1) After having checked once more for transitive linkages, we acknowledge your point. When including multiple intermediate linkages, all driving power and dependency power sums add up to 14. This has been corrected in figure 6.
(2) The MICMAC analysis is now presented as figure 3. Caption is placed accordingly.
Once again, thank you very much for taking your time to review the manuscript.
